# Intracellular uptake of macromolecules by brain lymphatic endothelial cells during zebrafish embryonic development

Max van Lessen[1,2,3], Shannon Shibata-Germanos[4], Andreas van Impel[1,2,3], Thomas A Hawkins[4], Jason Rihel[4], Stefan Schulte-Merker[1,2,3]*

[1]Institute of Cardiovascular Organogenesis and Regeneration, WWU Münster, Münster, Germany; [2]Faculty of Medicine, WWU Münster, Münster, Germany; [3]Cells-in-Motion Cluster of Excellence, WWU Münster, Münster, Germany; [4]Department of Cell and Developmental Biology, University College London, London, United Kingdom

**Abstract** The lymphatic system controls fluid homeostasis and the clearance of macromolecules from interstitial compartments. In mammals brain lymphatics were only recently discovered, with significant implications for physiology and disease. We examined zebrafish for the presence of brain lymphatics and found loosely connected endothelial cells with lymphatic molecular signature covering parts of the brain without forming endothelial tubular structures. These brain lymphatic endothelial cells (BLECs) derive from venous endothelium, are distinct from macrophages, and are sensitive to loss of Vegfc. BLECs endocytose macromolecules in a selective manner, which can be blocked by injection of mannose receptor ligands. This first report on brain lymphatic endothelial cells in a vertebrate embryo identifies cells with unique features, including the uptake of macromolecules at a single cell level. Future studies will address whether this represents an uptake mechanism that is conserved in mammals and how these cells affect functions of the embryonic and adult brain.

*For correspondence: schultes@ukmuenster.de

**Competing interests:** The authors declare that no competing interests exist.

## Introduction

Lymphatic vessels are present in most organs of the vertebrate body and serve in the uptake of macromolecules and excess fluid from interstitial space. Until recently both the eye and the brain had been considered to be immune privileged and devoid of lymphatics. This notion has been challenged by the discovery of the Schlemm's canal, a lymphatic-like vessel in the eye (*Aspelund et al., 2014*; *Park et al., 2014*; *Ramos et al., 2007*; *Thomson et al., 2014*), and by the identification of a lymphatic vascular network in the meningeal dura mater of the mouse brain (*Aspelund et al., 2015*; *Louveau et al., 2015*). While the concept of paravascular solute clearance from the brain into peripheral lymphatics had been proposed in previous decades (*Cserr and Ostrach, 1974*; *Yamada et al., 1991*), recent research into a sleep-dependent exchange between cerebral interstitial fluid and cerebrospinal fluid via astro-glial channels has revived interest in the concept. Coined the 'glymphatic' mechanism, this process would side-step the need for solutes in the brain to pass through the blood-brain endothelial barrier (*Iliff et al., 2012*; *Xie et al., 2013*). However, precisely how the cerebral interstitium and cerebrospinal fluid interconnect, as well as if and how the dural lymph vasculature participates in these perivascular clearance mechanisms remains enigmatic. Knowledge of how the energy intensive brain clears macromolecules and waste from its interstitial space will be critical to better understand not only the normal regulation of brain physiology and behaviour but also how pathological conditions such as neurodegenerative diseases may be affected

by improper clearance mechanisms (*Iliff et al., 2015*; *Tarasoff-Conway et al., 2015*; *Wostyn et al., 2015*). Moreover, although this system requires a complex interaction among blood endothelium, lymphatic structures and the brain, neither the developmental nor the evolutionary ontology of this system has been described.

The lymphatic vasculature of the peripheral nervous system consists of capillaries, which allow uptake of substances through specialized junctions, and larger collecting vessels, which transport the content of the vessel in a unidirectional manner towards the jugular vein, where lymph is finally funnelled into the vena cava. Lymphatic endothelial cells (LECs) have a molecular signature distinct from blood endothelial cells (BECs), including the expression of Prox1, Lyve1, and Podoplanin. Zebrafish transgenic reporter lines for *prox1a*, *flt4* (*van Impel et al., 2014*) and *lyve1* (*Okuda et al., 2012*) have allowed in vivo imaging of lymphangiogenic events in the trunk and the facial region of early embryos. There is a significant degree of conservation for lymphatic development on the genetic level between fish and mice, with mutants in the *ccbe1/vegfc/vegfr3* signalling axis all resulting in phenotypes lacking lymphatic structures (*Hogan et al., 2009a*, *2009b*; *Bos et al., 2011*; *Karkkainen et al., 2004*; *Le Guen et al., 2014*). In addition, both zebrafish *prox1a* maternal-zygotic mutants and *Prox1* mutant mice show lymphatic phenotypes (*Wigle and Oliver, 1999*; *Koltowska et al., 2015*). Whether zebrafish have brain lymphatics like mice has not been reported.

Here, we examine the development of brain lymphatics in the zebrafish embryo and find a pool of cells on the surface of the brain that display hallmark characteristics of LECs and yet do not form an endothelial sheet. These cells are positive for *prox1a*, *lyve1*, and *vegfr3* but express only low levels of the blood endothelial marker *kdr-like.* During later stages of development these cells populate the meningeal structures of the larval and adult brain. Functional assays based on tracer injections show that these cells take up exogenous substances similar to macrophages, and we provide evidence for an endocytic mechanism dependent on the mannose receptor (MR, Cluster of Differentiation 206, CD206) (*Martinez-Pomares, 2012*). However, unlike macrophages these cells are not of myelopoietic origin, suggesting that they constitute a unique cell type. The identification of brain lymphatic endothelial cells in an optically and experimentally tractable animal model complements existing efforts in the mouse to better understand the cellular components of a brain lymphatic system, their development, and their functionality.

## Results

### *flt4* positive cells sprout from the choroidal vascular plexus

Similar to mammals, meninges overlay the zebrafish brain (*Caruncho et al., 1993*). Recent studies in mice revealed the presence of lymphatic vessels in the dura mater, which function in macromolecule clearance (*Aspelund et al., 2015*; *Louveau et al., 2015*). To investigate zebrafish as a potential tool for the study of brain lymphatic development and function we analyzed *Tg(kdr-l:HRAS-mCherry-CAAX)$^{s916}$* (*Hogan et al., 2009a*); *Tg(flt4$^{BAC}$:mCitrine)$^{hu7135}$* (*van Impel et al., 2014*) double transgenic embryos (herein denoted as *Tg(kdr-l:mCherry);Tg(flt4:mCitrine)*). As *flt4* (*vegfr3*) is an established venous-lymphatic marker, and as *kdr-l* is expressed in all blood vessels, the simultaneous use of both markers distinguishes between lymphatic and blood endothelial cells (ECs). Before 56hpf there was no evidence of lymphatics in the embryonic brain (*Figure 1—figure supplement 1*). From around 56hpf however, *flt4* positive and low level *kdr-l* expressing cells sprout from a vessel proximal to the primary head sinus (PHS) and migrate along the mesencephalic vein (MsV) over the optic tectum (TeO) (*Figure 1A,B1–B7*, *Video 1*). Sprouting occurs from the choroidal vascular plexus (CVP) (*Figure 1C–C''*), and at 3dpf *flt4* positive cells form a bilateral loop of cells extending along the MsV over the brain surface (*Figure 1D,D'*).

### Sprouting endothelial cells express Prox1 and are sensitive to genetic ablation of *ccbe1* but not *pU.1*

The expression of *flt4* but not *kdr-l* in these venous-derived cells suggested a lymphatic nature. In *Tg(kdr-l:mCherry); Tg(flt4:mCitrine)* transgenics we observed that the putative LECs sprouting from the CVP contain residual mCherry protein indicative of trans-differentiation from the CVP (*Figure 1C–C''*). To confirm whether the same cells also express the lymphatic marker *prox1*, we carried out time-lapse imaging of the double transgenic line *Tg(prox1aBAC:KalTA4-4xUAS-E1b:*

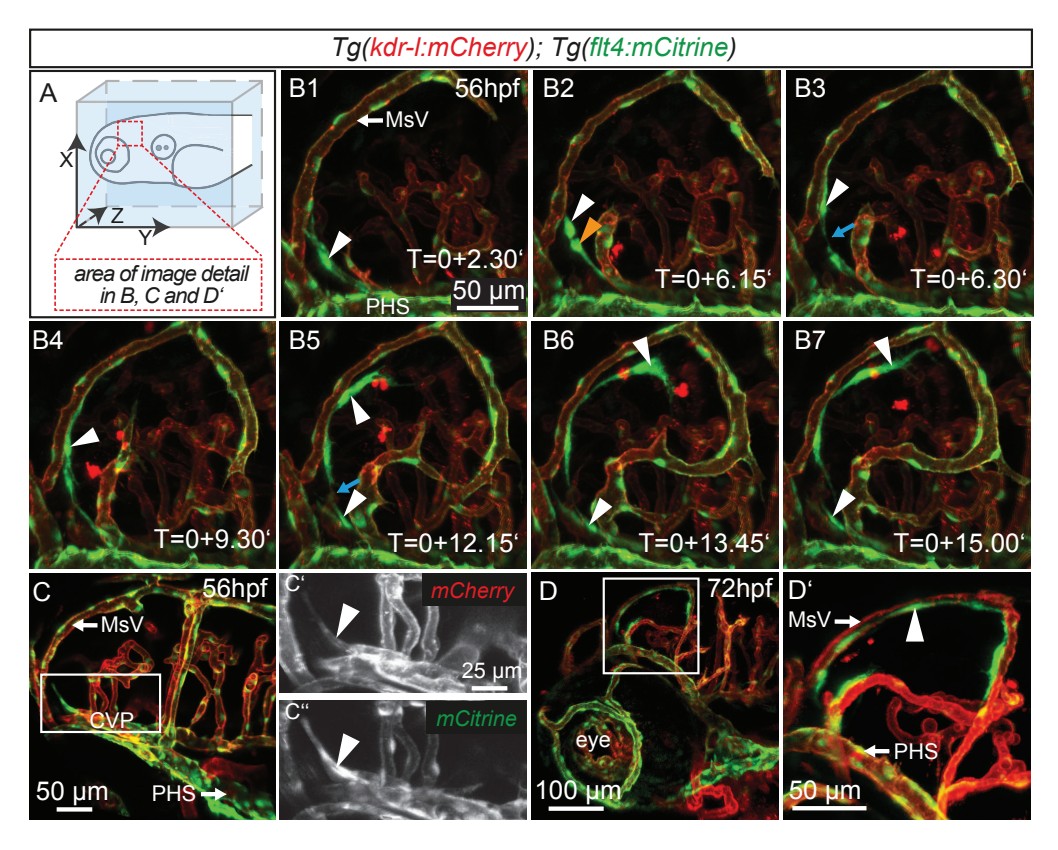

**Figure 1.** *flt4* positive cells sprout from the choroidal vascular plexus and migrate along blood vessels. In all images blood vessels are highlighted in red (*kdr-l:mCherry*) and lympho-venous cells in green (*flt4:mCitrine*). (**A**) Overview and orientation of zebrafish embryos imaged in **B–D**. (**B1 – B7**) Time-lapse still images of lateral confocal projections of the TeO. At 56hpf strong mCitrine positive but low mCherry expressing ECs sprout from a vessel behind the PHS and migrate along the MsV (white arrow). Following initial sprouting the cell divides (**B2**, white and orange arrowheads). Leading and following cells appear to temporarily lose contact (**B3** and **B5**, blue arrow). After making contact to the MsV the sprout continues migration (**B4–B7**). (**C**) Partial projection of the sprouting cells (cropping of the PHS) reveals that the migrating cells originate from the more proximal positioned CVP at around 56hpf. Sprouting cells express low mCherry but high mCitrine levels compared to the CVP (inset **C'–C''**). (**D**) Lateral confocal projection of the head region shows that at 72hpf *flt4* positive ECs (white arrowheads) form a loop aligned next to the MsVs (white arrowheads). (**D'**) Higher magnification of the boxed area in (**D**). Data are representative of at least five independent experiments. CVP, choroidal vascular plexus; EC, endothelial cell; hpf, hours post fertilization; MsV, mesencephalic vein; PHS, primary head sinus; TeO, Optic Tectum. Apostrophe in B1–B7 denotes hours.

The following figure supplement is available for figure 1:

**Figure supplement 1.** *flt4* positive cells develop between 56 and 72hpf.

uncTagRFP)[nim5] (*van Impel et al., 2014*) (herein denoted *Tg(prox1a:KalTA4, UAS:TagRFP)*); *Tg(flt4:mCitrine)* at 56hpf. Indeed, *flt4* positive sprouts show *prox1a* expression, and this expression persists in 5dpf embryos (*Figure 2A–D*). We corroborated *prox1* expression observed in the reporter line by antibody staining and found Prox1 positive nuclei both in lymphatic sprouts and the CVP at 56hpf (see Figure 4A–B''). In addition, analysis of *Tg(lyve1:DsRed2)[nz101]*(*Okuda et al., 2012*); *Tg(flt4:mCitrine)* 5dpf embryos revealed the expression of another lymphatic marker, *lyve1* (*Figure 2E–E''*).

While expression of *flt4*, *prox1a* and *lyve1* are strong indicators of endothelial and lymphatic identity, we still wanted to evaluate whether these cells might belong to the macrophage lineage. Analysis of the double transgenic *Tg(lyve1:DsRed2)[nz101]*; *Tg(mpeg1:EGFP)[gl22]*(*Ellett et al., 2011*) embryos revealed that *mpeg1* positive macrophages (arrows in *Figure 3A'–A'''*) are distinct from *lyve1* positive cells (arrowheads in *Figure 3A'–A'''*). Only at very high exposure levels could we detect low

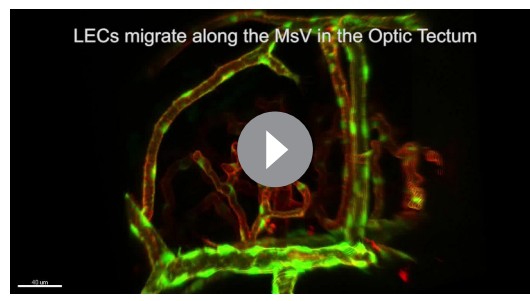

**Video 1.** LECs migrate along the MsV in the Optic Tectum. Complete time-lapse of partial lateral confocal projections of the zebrafish TeO from 56hpf to 72hpf as depicted in *Figure 1*, B1-B7. Blood vessels are highlighted in red (*kdr-l:mCherry*) and lympho-venous cells in green (*flt4:mCitrine*). Data are representative of at least five independent experiments. hpf, hours post fertilization; LEC, lymphatic endothelial cell; TeO, Optic Tectum.

amounts of *mpeg1* expression in *lyve1* positive cells (*Figure 3A'''*). In addition, inhibiting myelopoiesis by injecting a *pU.1* (*spi1b*) morpholino at the one-cell stage (*Rhodes et al., 2005*) efficiently ablated *mpeg* positive macrophages but did not block the development of the *lyve1* positive cells close to the MsV (*Figure 3B–B'''*).

The formation of lymphatic structures in fish and mice are dependent on Vegfc/Vegfr3 signaling (*Karkkainen et al., 2004*; *Küchler et al., 2006*; *Yaniv et al., 2006*) and Ccbe1 activity (*Hogan et al., 2009a*; *Bos et al., 2011*). To investigate if Vegfc signaling not only influences lymphangiogenesis in the trunk (*Hogan et al., 2009b*) but also head lymphatic development, we analyzed *vegfc*^hu6410^ (*Helker et al., 2013*) and *ccbe1*^hu10965^ (*Kok et al., 2015*); Tg(flt4: mCitrine); Tg(flt1^enh^:tdTomato) ^hu5333^ (*Bussmann et al., 2010*) fish, which do not express functional Vegfc and Ccbe1, respectively. As expected for lymphatic structures, *ccbe1* mutant embryos (and *vegfc* mutants, data not shown) but not sibling controls completely lack these cells, as assessed by either Prox1 antibody staining or *flt4* expression (*Figure 4A–E*). In situ hybridization for *vegfc* and *ccbe1* in wild type embryos did not reveal detectable up-regulation of either gene in the immediate vicinity of the MsV (data not shown).

Taken together, the cells sprouting from the CVP express the *bona fide* lymphatic markers *flt4*, *lyve1*, and *prox1*. The blood vascular-marker *kdr-l* and the macrophage marker *mpeg1* are expressed only at very low levels. Ablation of the myelopoietic lineage eradicates macrophages but not the CVP derived cells. In addition, Vegfc signaling, the main lymphangiogenic pathway in other tissue beds, also governs development of these cells. This is strong evidence of a lymphatic identity, and we therefore term this cell population as brain lymphatic endothelial cells (BLECs).

## BLECs cover the larval and adult brain

Time lapse movies of the development of these cells and inspection of 3D-rotations of 5dpf zebrafish demonstrate that BLECs do not penetrate deeper into the brain (*Figure 5A,B*, *Video 2*). During larval stages BLECs cover the distal periphery of the optic tectum, and many are positioned in close proximity to meningeal blood vessels (*Figure 5C,D*, *Video 3*). *flt4* positive BLECs remain into adulthood, taking on a diffuse pattern across most outer surfaces of the brain. BLECs have distinct organisation and morphology according to the specific brain surface they occupy (*Figure 5E–J*). The telencephalon is nearly devoid of *flt4:mCitrine* signal on the dorsal surface but a few positive cells cluster around arterioles that branch from the anterior cerebral carotid artery at the lateral midline (*Figure 5F*). Some of these BLECs stretch dorso-laterally along the vasculature (*Figure 5J*). In contrast, tectal BLECs are dense and extend long thin processes to form close associations with one another and with the *kdr-l* positive vessels (*Figure 5G*, arrowheads). Cerebellar BLECs are less dense and exhibit an elongated and flattened morphology that stretches along the vasculature (*Figure 5H*). A striking feature of these cells are spherical internal compartments resembling endosomes or vacuoles (*Figure 5G,H*). Coronal sections of the adult zebrafish optic tectum reveal *flt4* positive BLECs are closely associated with surface vessels of the brain, remaining limited to the meningeal space without extending further into the neuropil (*Figure 5K*).

## BLECs have characteristic ultrastructures that are distinct from pericytes

To determine whether BLECs have distinguishing ultrastructural features, samples of adult zebrafish brains were prepared for transmission electron microscopy (TEM). Within the tectal and

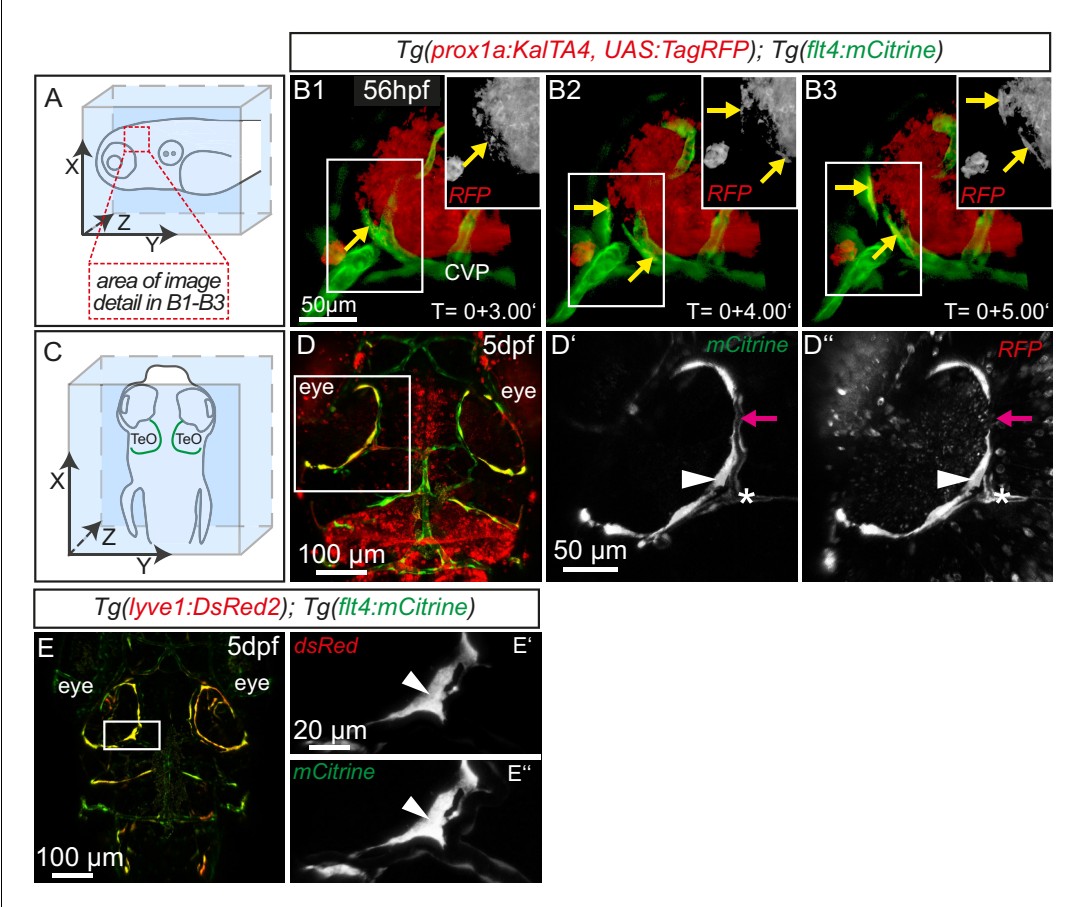

**Figure 2.** Sprouting *flt4* positive endothelial cells express *prox1a* and *lyve1*. (**A**) and (**C**) Overview and orientation of 56hpf and 5dpf zebrafish embryos shown in B1-B3 (**A**) and **D–E''** (**C**). Green lines in **C** illustrate the position of mCitrine positive ECs. (**B1–B3**) Time-lapse still images of partial lateral confocal projections of the TeO as depicted in **A** (red inset). At 56hpf tagRFP indicates *prox1a* promoter activity in mCitrine positive endothelial sprouts (yellow arrows). Note that the RFP positive red cell mass above the CVP likely reports prox1a expression in cranial ganglia within the TeO. (**D**) Strong *prox1a* promoter activity persists in the mCitrine positive loop structure (inset, white arrowhead) but is absent in neighboring MsV (inset, pink arrow) at 5dpf. Note also tagRFP expression around a small part of the MsV (inset, white asterisk). (**E**) *flt4* positive ECs express *lyve1* at 5dpf (**E',E''** inset, white arrowhead). Data are representative of at least three independent experiments. CVP, choroidal vascular plexus; dpf, days post fertilization; hpf, hours post fertilization; MsV, mesencephalic vein; TeO, Optic Tectum. Apostrophe in **B1–B3** denotes hours.

telencephalic meninges, we identified a population of cells near blood vessels that possess large, circular, electron dense vesicles in their cytoplasm (*Figure 6A–C*). The location near vessels, pyramidal shape, and frequency of these cells are consistent with this cell population being identical to BLECs. The diameters of the large electron dense inclusions in ultrathin TEM slices can be larger than 2 μm, consistent with vesicle measurements taken from confocal stacks of the tectum (1.96 ± 0.04 μm, mean ± SEM, n = 210 vesicles). Other neighboring cells have smaller, heterogeneous, irregularly-shaped electron dense and pale inclusions, features that are consistent with Mato or fluorescent granular perithelial cells (FGPs) (*Figure 6A*) (*Mato et al., 2009*). BLECs were not found within the basement membrane surrounding meningeal blood vessels, and even in examples that were closely associated with vasculature, cellular processes or somata intervened between the BLEC and ECs of blood vessels (*Figure 6A',B',C*). In contrast, pericytes maintained intimate contact with ECs in both meningeal and pial brain tissue (*Figure 6C,D*), lying within the basement membrane of the vasculature (*Figure 6D*). TEM sections of tectal and telencephalic meninges revealed that BLECs are located in the highly interdigitated inner layer (*Momose et al., 1988*; *Wang et al., 1995*) of the meninges, consistent with a close association with blood vessels (*Figure 6E,F*).

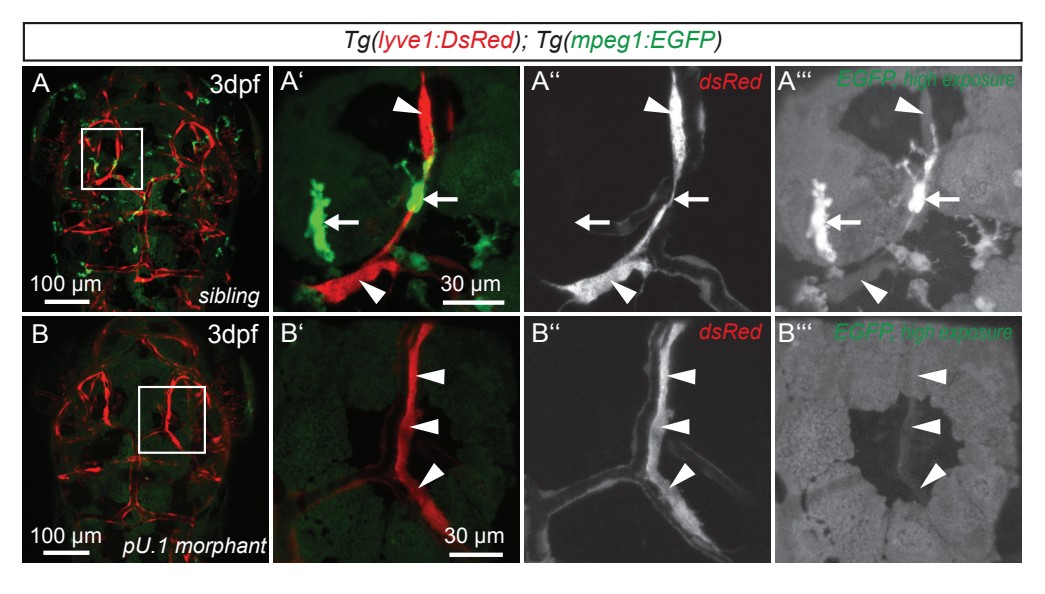

**Figure 3.** *flt4* positive cells are not of myelopoietic origin. (**A**) and (**B**) Dorsal view of partial confocal projections of 3dpf double transgenic *Tg(lyve1: DsRed)^{nz101}; Tg(mpeg1:EGFP)^{gl22}* embryos. Brain resident macrophages strongly express *mpeg1:EGFP* (**A'–A'''**, white arrows) while *lyve1* positive LECs are only weakly *EGFP* positive (**A'–A'''**, white arrowheads) in uninjected control embryos. Depletion of the myelopoietic lineage by injection of *pU.1* (*spi1b*) morpholinos ablates all *EGFP* positive macrophages but does not affect the formation of *lyve1* positive LECs (**B'–B'''**, white arrowheads). dpf, days post fertilization; LEC, lymphatic endothelial cell.

## BLECs take up macromolecules

Lymphatics in other tissue beds across species form lumenized vessels and ultimately drain into the venous circulation (*Schulte-Merker et al., 2011*; *Yang and Oliver, 2014*). In contrast, BLECs in fish present as a loose network of cells both at embryonic (*Figure 1B3*) and adult (*Figure 5H*) stages, with no apparent organization as a lumenized endothelium, and consequently no connection to the blood circulation (*Figure 7—figure supplement 1*). Nevertheless, in order to address a possible functional role of BLECs in fluid drainage or macromolecule clearance, we performed intracerebral injection of different fluorescent tracers into the TeO (*Figure 7A*). Strikingly, injection of a 150 kDa IgG-conjugated Alexa Fluor 674 (IgG-647) accumulated at all *flt4* positive BLECs in the head of 5dpf embryos but not at neighboring *kdr-l* positive blood vessels (*Figure 7B–B'*). Intriguingly, time-lapse analysis after IgG-647 injection revealed dynamic movement of tracer in BLECs (*Video 4*). Analysis of uninjected controls revealed auto-fluorescent signal from the eyes and skin pigmentation (*Figure 7— figure supplement 2*).

To investigate if BLECs can also take up macromolecules from the ventricles, we injected IgG-647 directly into the hindbrain ventricle in 3dpf fish. Time-lapse imaging revealed accumulation of tracer in BLECs over time (*Figure 7—figure supplement 3*, *Video 5*).

Next, we addressed if molecular weight affects tracer uptake by injecting different MW dextrans in the TeO. We observed a crisp tracer signal at BLECs with little background noise after injection of 10 kDa dextran-conjugated Alexa Fluor 647, similar to 150 kDa IgG-647 (*Figure 7C,D*). Conversely, the uptake of a 500 kDa Cy5-dextran was weak and no uptake could be observed for 2000 kDa FITC-dextran (*Figure 7E,F*). Similar to 2000 kDa FITC-dextran, we also found no uptake of the 960 Dalton small molecule dye Evans Blue (*Figure 7G*). We conclude that BLECs can efficiently and selectively take up polysaccharides and glycoproteins ranging at least from 10 to 150 kDa, and possibly larger.

## BLECs internalize cargo through endocytosis

Subsequently, we addressed if the tracers only attach to the plasma membrane (PM) or if macromolecules are internalized by BLECs. Following internalization, cargo typically enters endosomal sorting

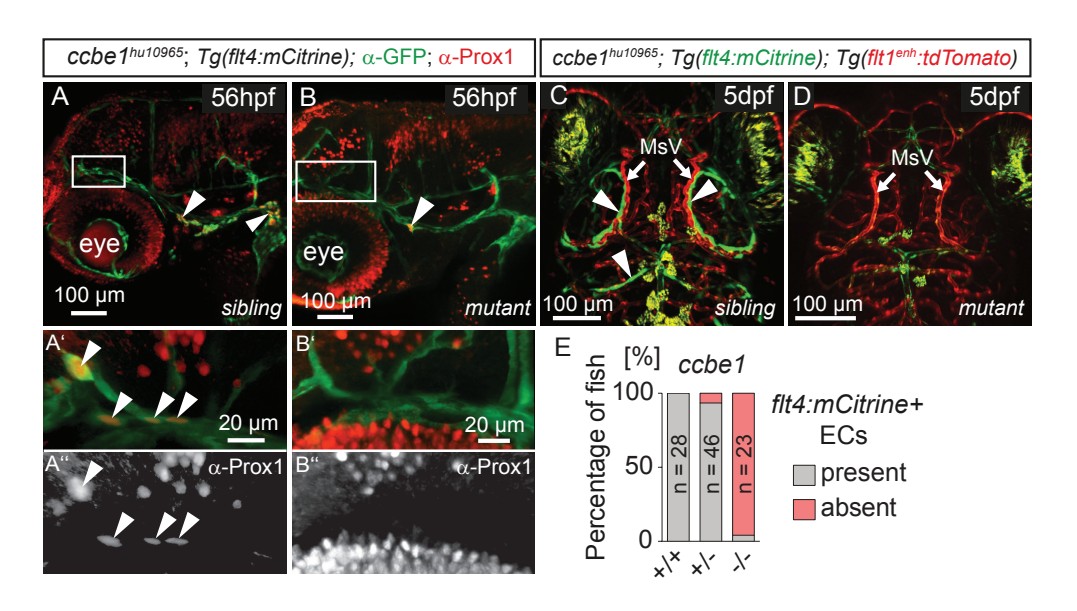

**Figure 4.** *flt4* positive cells are sensitive to ablation of the *vegfc/ccbe1* pathway. (**A**) and (**B**) Antibody staining detects Prox1 protein in the GFP positive CVP and sprouting ECs in wild type sibling (**A**) but not in *ccbe1* mutant (**B**) embryos (**A',A''** insets, white arrowheads). Note few Prox1 expressing cells remaining in mutants at the site of facial lymphatic sprouting (**B**, white arrowhead). (**C**) and (**D**) Absence of *flt4:mCitrine* expressing ECs in *ccbe1* mutant (**D**) but not in sibling controls (**C**, white arrowheads) in 5dpf embryos. *Tg(flt1enh:tdTomato)* labels arteries. (**E**) Quantification of results shown in **C** and **D**. Data are representative of at least two independent experiments. CVP, choroidal vascular plexus; dpf, days post fertilization; EC, endothelial cell; hpf, hours post fertilization; MsV, mesencephalic vein; +/+, wildtype; +/-, heterozygous; -/-, mutant.

The following source data is available for figure 4:

**Source data 1.** Zebrafish embryos from an incross of *Tg(flt1enh:tdTomato); Tg(flt4:mCitrine)* heterozygous for the mutant *ccbe1hu10965* allele were analyzed by fluorescent stereomicroscopy at 5dpf for the presence or absence of *flt4:mCitrine* expressing ECs in the head (as depicted in **Figure 4C and D**).

for lysosomal degradation (**McMahon and Boucrot, 2011**). En route to the lysosomal compartment, endocytic vesicles gradually acidify down to a pH of 4.6–5.0 (**Mellman et al., 1986**). Thus, exploiting this pH-shift, we injected pHrodo Red Avidin (pHr), which strongly fluoresces at low pH, and observed robust uptake into endocytic vesicles (**Figure 7H**, **Video 6**). To further corroborate the uptake of pHr we also injected pHr along with IgG-647 into the ventricle. Again, both tracers clearly label BLECs; however, only IgG-647 but not pHr fluoresces at neutral pH, for example in the ventricles (**Figure 7—figure supplement 4**). Therefore, we conclude that BLECs internalize the fluorescent tracers and traffic them to more acidic endosomal compartments.

To follow the internalized cargo in BLECs in a dynamic manner, we injected embryos with purified KAEDE protein, which efficiently labeled BLECs (**Figure 7I**) in the TeO. Exposure to UV-light irreversibly converts KAEDE from green to red fluorescence (**Ando et al., 2002**). To minimize conversion of free extracellular KAEDE, we converted BLECs only in a small region of interest (**Figure 7I–I'''**). Subsequent time-lapse imaging showed rapid disappearance of red fluorescence after 10 min, which was hardly detectable after 80 min. Over this period, BLECs re-internalized readily detectable unconverted green KAEDE from the interstitial space, as seen by the re-appearance of unconverted, green KAEDE molecules in BLECs (**Figure 7I–J'''**).

The small size of the vesicles observed in BLECs after tracer injection suggests an endocytic uptake mechanism (**Ganley et al., 2004**), although other routes for internalization are possible. To address whether endocytosis is the mechanism underlying cargo uptake into BLECs, we first treated 5dpf embryos with the dynamin inhibitor Pyrimidyn-7 (**McGeachie et al., 2013**) and then injected the pH-sensitive pHr and the IgG-647 tracer dyes. In DMSO control treated embryos, dye injection into the TeO resulted in efficient uptake of both dyes into BLECs and co-localization in the same

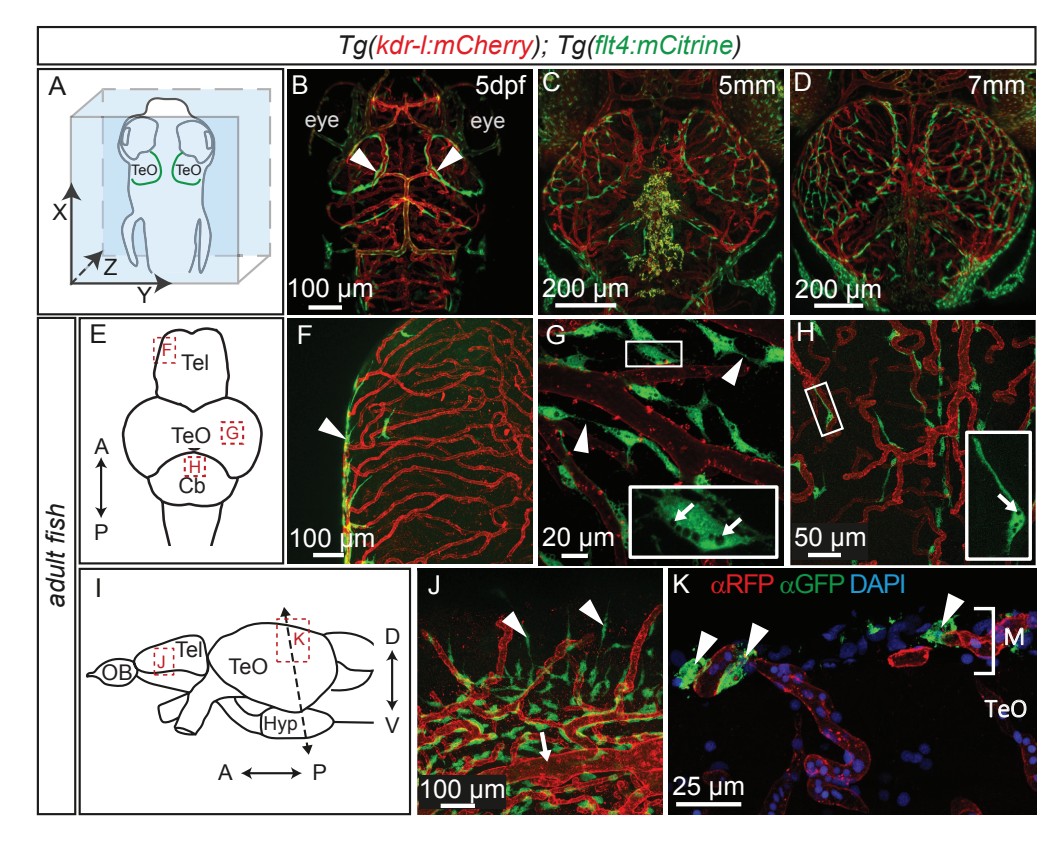

**Figure 5.** Perivascular position of *flt4* positive BLECs in larval stages and adult meninges. (**A**) Overview and orientation of zebrafish embryos imaged in **B–D**. (**B**) Dorsal confocal projections of the head region show that at 5dpf *flt4* positive BLECs (white arrowheads) form a bilateral loop in the TeO which aligns next to the MsVs (white arrowheads). (**C**) and (**D**) Dorsal confocal projections at 5 mm and 7 mm stages show increasing spreading of *flt4* positive ECs, particularly over the TeO. (**E**) Schematic diagram (dorsal view) of adult brain. Dotted boxes show area of image detail of **F–H**. (**F**) The dorsal surface of the Tel displays a distinct absence of *flt4:mCitrine* positive cells. (**G**) and (**H**) A dorsal view of the TeO and Cb, respectively, shows *flt4: mCitrine* positive BLECs (green) closely associated with surface *kdr-l:mCherry* positive blood vasculature (red), interacting with vessels and one another via thin processes (white arrowheads). BLECs contain multiple internal compartments (inset, white arrows). BLECs on the cerebellar surface are less dense and are morphologically distinct, exhibiting an elongated shape (inset). (**I**) Schematic diagram (lateral view) of adult brain. Dotted boxes show area of image detail of **J** and **K**, and the dotted line shows the section plane in **K**. (**J**) BLECs on the lower half of the lateral telencephalic midline boundary cluster around smaller vessels branching off from the anterior cerebral carotid artery at the lateral midline of the telencephalon (white arrow) with a few cells sending sparse and elongated processes towards the dorsal surface (white arrowheads). (**K**) Coronal section of the adult brain shows *flt4:mCitrine* positive BLECs (green) (white arrowheads) closely associated with *kdr-l:mCherry* positive vasculature (red) exclusively within the meninges (**M**) but not along the vessels extending ventrally into the tectal neuropil. DAPI nuclear counter stain (blue). Data are representative of at least three independent experiments. BLEC, brain lymphatic endothelial cell; Cb, Cerebellum; dpf, days post fertilization; EC, endothelial cell; Hyp, hypothalamus; M, Meninges; MsV, mesencephalic vein; Tel, Telencephalon; TeO, Optic Tectum.

endocytic vesicles (*Figure 8A,B*). Conversely, treatment with Pyrimidyn-7 almost completely blocked pHr endocytosis (*Figure 8C*) demonstrating that internalization is dynamin dependent.

In mammals, the mannose receptor (MR) serves as a pattern recognition receptor (PRR) in the uptake of various macromolecules through endocytosis (*Martinez-Pomares, 2012*). The zebrafish genome encodes two genes of the mannose receptor, *mrc1a* and *mrc1b* and in situ hybridization revealed specific expression of *mrc1a* in BLECs in the meninges (*Figure 8—figure supplement 1*). Although MR expression in mouse lymphatics has thus far been implicated only in the homing of T-cells and tumor metastasis (*Marttila-Ichihara et al., 2008*), we decided to examine whether BLECs use the MR to internalize macromolecules. To test this, we examined the inhibitory effect of mannan, a bacterial polysaccharide that binds with high affinity to the MR (*Sallusto et al., 1995*), on cargo uptake into BLECs. An initial control injection of bovine serum albumin (BSA) followed by pHr and

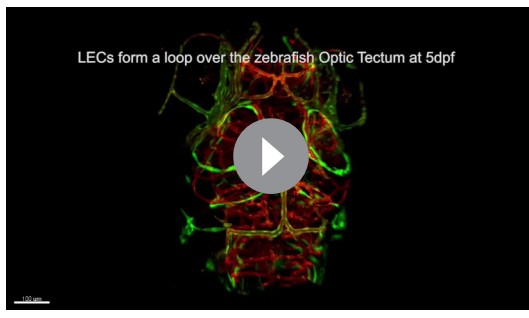

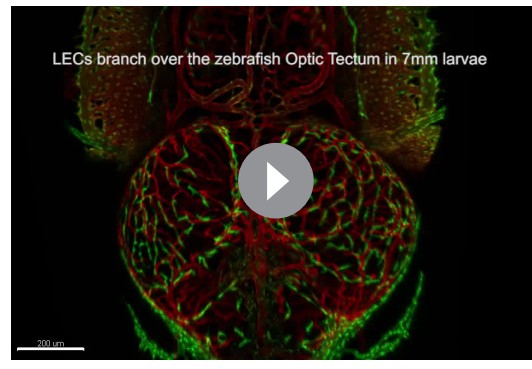

**Video 2.** LECs form a loop over the zebrafish Optic Tectum at 5dpf. 3D rotation of a dorsal confocal projection of the zebrafish head region of a 5dpf embryo with blood vessels highlighted in red (*kdr-l: mCherry*) and lympho-venous cells in green (*flt4: mCitrine*). The *flt4* positive LECs form a loop over the TeO and appear not to penetrate deeper into the brain. Halfway through the video a 3D surface rendering has been included marking *kdr-l* positive vessels in red and cells which strongly express *flt4: mCitrine* in green. Data is representative of at least five independent experiments. dpf, days post fertilization; LEC, lymphatic endothelial cell; TeO, Optic Tectum.

**Video 3.** LECs branch over the zebrafish Optic Tectum in 7 mm larvae. 3D animation of a dorsal confocal projection of the zebrafish head region of a 7 mm larva with blood vessels are highlighted in red (*kdr-l: mCherry*) and lympho-venous cells in green (*flt4: mCitrine*). *flt4* positive LECs cover the TeO and appear not to penetrate deeper into the brain. Many LECs are in close association with *kdr-l* positive blood vessels. Also note the isolated LECs with no apparent contact to other LECs or blood vessels. Data is representative of at least three independent experiments. LEC, lymphatic endothelial cell; TeO, Optic Tectum.

IgG-647 resulted in efficient uptake and co-localization of both dyes in BLECs similar to DMSO treated embryos (*Figure 8D*). Similarly, injection of the negative control galactose, which does not inhibit MR uptake, had no effect on the ability of BLECs to internalize either pHr or IgG-647 (*Figure 8—figure supplement 2*). However, an initial injection of mannan completely stopped subsequent pHr uptake, while IgG-647 association with BLECs remained relatively unperturbed (*Figure 8E*). We hypothesized that following the mannan treatment, the tracers may be accumulating on the PM of BLECs without internalizing. To test this, we predicted that injecting mannan after pHr but before the IgG-647 would allow for the internalization of the pH-sensitive dye, while the IgG-647 would subsequently not internalize. Consistent with this hypothesis, both dyes accumulated at BLECs but did not co-localize in internal compartments (*Figure 8F*). Since the dynamic movement of tracer dyes in BLECs impeded high spatial resolution imaging in live embryos (*Video 4*), we repeated the mannan injection series followed by PFA fixation. Confirming our live imaging data, 3D image reconstruction revealed co-localization of pHr and IgG-647 inside BLECs in BSA control treated embryos (*Figure 8G'*). In contrast, upon mannan inhibition, the IgG-647 appears stuck at the level of the PM (*Figure 8H',I'*).

## Discussion

While investigating the possible presence of lymphatic vessels in the embryonic and adult zebrafish brain, we have identified a unique population of cells that exhibit many hallmark characteristics of LECs and which we named *brain lymphatic endothelial cells*, or BLECs. These cells appear to be only loosely connected and do not form endothelial monolayers. Their ability to take up exogenous substances is reminiscent of macrophage and dendritic cell activity (*Steinman et al., 1976*; *Platt et al., 2010*), while their proximity to blood vessels suggest a possible perivascular cell lineage (*Sweeney et al., 2016*). However, there are a number of criteria which makes these cells unlikely to belong to either of these latter lineages.

BLECs sprout from a specific vessel, the CVP, at 56hpf. Concomitant with sprouting, they downregulate the BEC marker *kdr-l* while, conversely, *flt4* becomes up-regulated. This expression profile is common for lymphatic ECs (*Bussmann et al., 2010*). Consistent with this notion, the LEC markers

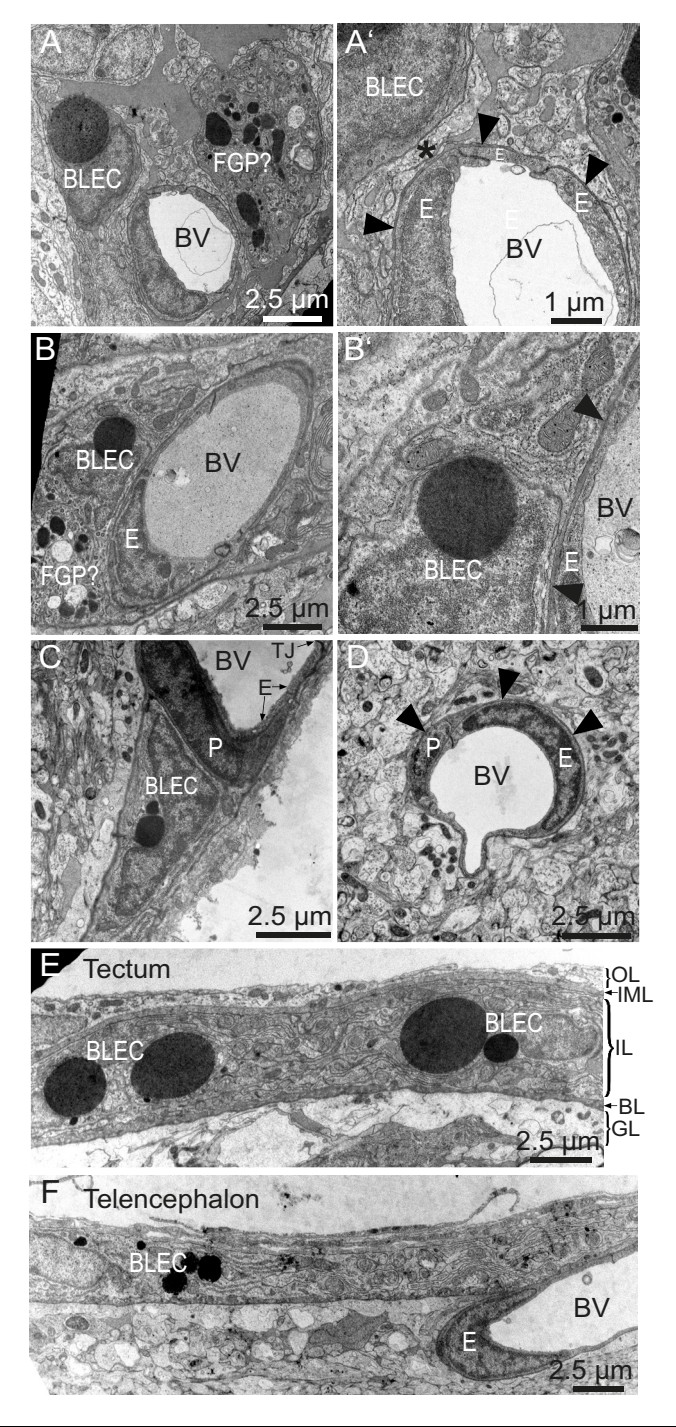

**Figure 6.** BLEC ultrastructure revealed in TEM of adult brain meninges. (**A**) and (**B**) TEM micrographs of tangential sections of the domed tectal surface revealed cells (BLEC) with large spherical inclusions in proximity to meningeal blood vessels (BV). Higher magnification shows that BLECs are not contained within the basement membrane (**A'** and **B'**, arrowheads) of endothelial cells (**E**) and are separated from the vessels by other cellular processes (e.g. asterisk in **A'**). Putative Mato/FGP cells are indicated (FGP?). (**C**) and (**D**) Pericytes (P) maintain close contact with endothelial cells (**E**) and are found under the basement membrane (arrowheads). BLECs are only found in the meninges (**C**) and are absent near blood vessels in the neuropil (**D**) of the brain. (**E**) and (**F**) T/S sections of tectal (**E**) and latero-ventral telencephalic (**F**) meninges showing BLECs are present in the inner layer (IL) of the meningeal covering. BLEC, brain lymphatic endothelial cell; **E**, endothelial cell; FGP, fluorescent granular
*Figure 6 continued on next page*

*Figure 6 continued*

perithelial cell; P, pericyte; TJ, tight junction. Nomenclature of meningeal layers according to *Momose et al. (1988)*: OL, outer layer; IML, intermediate layer; IL, inner layer, BL, basal lamina; GL, glia limitans.

*prox1a* (*van Impel et al., 2014*; *Koltowska et al., 2015*) and *lyve1* (*Okuda et al., 2012*) are up-regulated in these cells as soon as they sprout from the CVP. Since lymphangiogenesis is exquisitely sensitive to Vegfc/Vegfr3 signaling (*Hogan et al., 2009b*; *Karkkainen et al., 2004*), the complete loss of these LECs in the brain of *ccbe1* (*Figure 2G–K*) and *vegfc* mutants (data not shown) also argues for a lymphatic lineage (*Hogan et al., 2009a*). The sensitivity to Vegfc signaling, the expression of vascular markers, and the direct endothelial origin from the CVP all exclude pericytes or other non-vascular perivascular cells. Ultrastructural analysis further argues against pericytes, as basement membranes shared with ECs were not found in the case of BLECs.

This newly identified population of cells therefore shares many features with LECs while never aggregating to form lumenized vessels (*Figure 5G–K*). In this respect BLECs are different from venous-derived parachordal lymphangioblasts in the zebrafish embryonic trunk, which experience an intermediate, single cell phase but then go on to form the thoracic duct and the dorsal longitudinal lymphatic vessel (*Hogan et al., 2009a*; *Padberg et al., 2017*). There are few other examples of endothelial cells not belonging to a contiguous epithelium, most notably liver sinusoidal endothelial cells (LSCEs) (*Braet and Wisse, 2002*).

Intriguingly, the high capacity endocytosis and degradation of extracellular material observed in LSCEs is also dependent on the mannose receptor (MR), *Mrc1*, in mice (*Elvevold et al., 2008*). High *Mrc1* expression is also a distinguishing feature of perivascular macrophages (pvMΦ) (*Elvevold et al., 2008*), which strongly resemble BLECs in their morphology and close localization to blood vessels in the murine CNS. However, unlike BLECs, pvMΦ derive from hematopoietic precursors (*Goldmann et al., 2016*), and sprouting of macrophages directly from an endothelial source has not been described (*Epelman et al., 2014*). In addition, we monitored CNS resident macrophages by their expression of the reliable zebrafish macrophage marker *mpeg1* (*Ellett et al., 2011*), and this expression was extremely low in *lyve1* expressing BLECs (*Figure 3B*) but strong and robust in adjacent tissue resident macrophages (*Figure 3A'*). Furthermore, BLECs remained intact when the myelopoietic lineage was deleted (*Figure 3*). In mice, macrophages express *Lyve1* during early embryonic stages but expression does not persist in the adult (*Bos et al., 2011*), while we observe *lyve1* expression in BLECs of adult fish (data not shown). Hence, although BLECs resemble pvMΦ in morphology and function we could not find further supporting evidence for these cells to be part of the macrophage lineage. Even though the expression of *flt4* and *prox1a* and the virtual absence of *mpeg1* expression is a strong indicator for a lymphatic fate, marker expression in and of itself has to be considered with some caution when categorizing cells. Therefore, we consider the direct derivation from an endothelial structure (CVP, *Figure 1B–C*) and the presence of BLECs in *pU.1* morphants to be the strongest evidence that BLECs indeed belong to an endothelial and not to a macrophage lineage.

Further, we followed the distribution of BLECs over time, through larval and adult stages. BLECs often are in close proximity to blood vessels, possibly also suggesting a functional interconnectivity. Indeed, BLECs were recently found to express pro-angiogenic factors such as *vegfab* and *egfl7* (*Bower et al., 2017*; *Venero Galanternik et al., 2017*) required by blood vessels in the brain. Furthermore, genetic and laser-ablation mediated depletion of BLECs results in a concomitant reduction of blood vessels, and recovery of BLECs precedes regeneration of blood vessels in the affected areas during larval stages (*Bower et al., 2017*). Anatomically, BLECs cover many, but not all, surfaces of the brain, and display distinct morphological characteristics depending on which region of the brain they occupy. For example, the dorsal surface of the adult telencephalon completely lacks BLECs despite having many blood vessels deeper in the brain. This likely reflects the everted brain development in teleosts, the consequences of which make the dorsal telencephalon a ventricular surface. Functional studies in these late stages after skull formation are difficult but will be required to examine whether morphological differences reflect possible functional distinctions across brain regions.

A particularly intriguing feature of BLECs in the optic tectum and other parts of the brain is the presence of many large, spherical and electron dense vacuoles or granules in adult fish (*Figures 5G,H* and *6*). These are likely a direct indication of endocytic or phagocytic activity of BLECs and are reminiscent

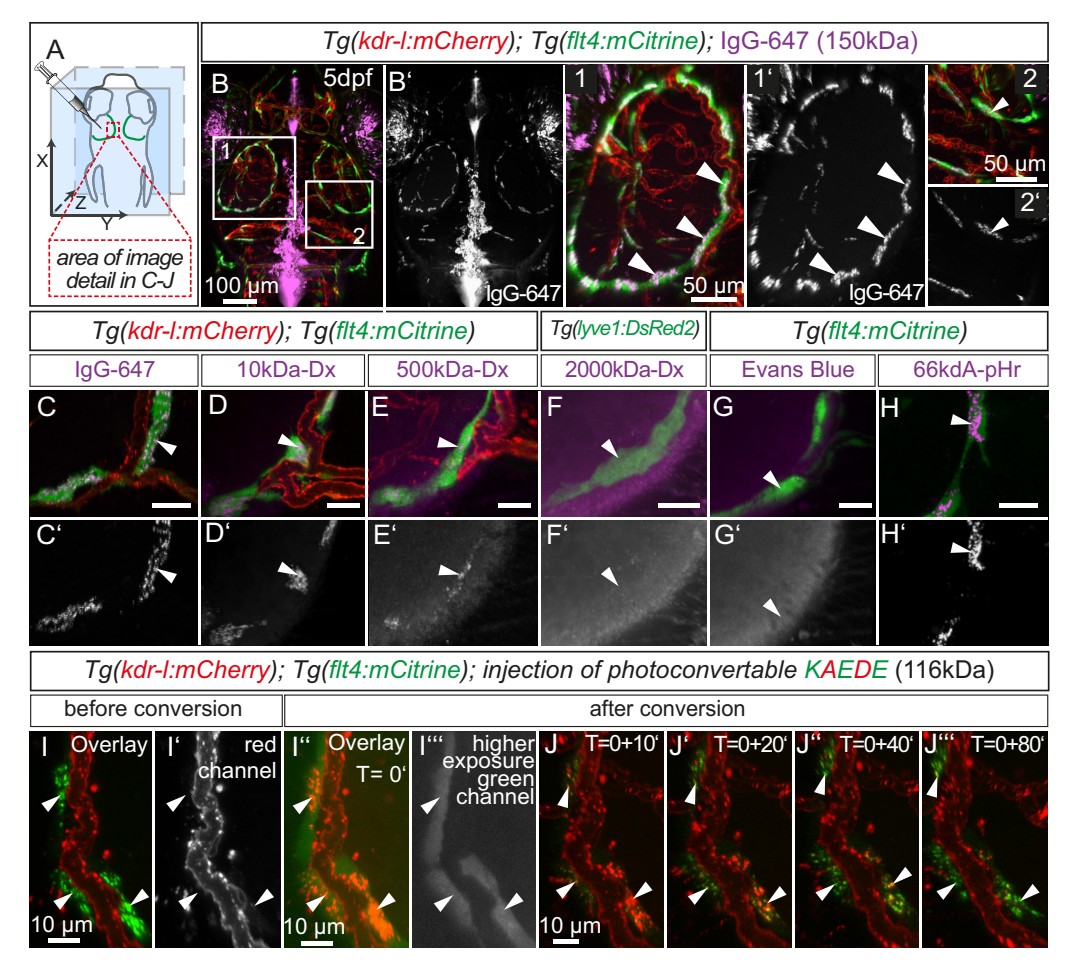

**Figure 7.** BLECs dynamically take up macromolecules. (**A**) Overview of the zebrafish head region and intratectal injection site of fluorescent dyes into the center of the TeO close to the meninges. Red inset denotes area of image detail in **C–J**. (**B–J**) Representative dorsal confocal projection of 5dpf embryos injected with different tracer dyes as indicated. Injected IgG-647, 10kDa-Dx, 500kDa-Dx and 66kDa-pHr specifically localize to vesicles in mCitrine positive BLECs but not to mCherry positive blood vessels (**B–E**, **H**, **I** and **J**, white arrowheads). BLECs do not accumulate high MW dextran or Evans Blue (**F** and **G**, white arrowhead). BLECs strongly label with the pH-sensitive dye pHrodo (pHr) indicating tracer internalization (**H**, white arrowhead). (**I**) and (**J**) Injected KAEDE protein is taken up by BLECs (**I** and **I'**, white arrowheads) and can be photoconverted to red by localized UV-exposure to a region of interest (**I''** and **I'''**, white arrowheads). 10 min after photoconversion, the amount of converted red KAEDE decreases dramatically (**J**) and the signal is almost completely lost after 80 min (**J'''**, white arrowheads). In the same time course photoconverted BLECs re-accumulate non-converted green KAEDE protein (**J'''**, white arrowheads). Scale bar in **C–H** corresponds to 20 μm. Data are representative of at least two independent experiments. BLEC, brain lymphatic endothelial cell; dpf, days post fertilization; Dx, Dextran; IgG-647, IgG-conjugated Alexa Fluor 674; TeO, Optic Tectum; pHr, pHrodo Red Avidin. Apostrophe in **J** denotes minutes.

The following figure supplements are available for figure 7:

**Figure supplement 1.** BLECs do not carry blood flow.

**Figure supplement 2.** Eyes and skin pigmentation is autofluorescent in non-injected embryos.

**Figure supplement 3.** BLECs take up tracer from ventricles.

**Figure supplement 4.** pH-sensitive pHrodo Red Avidin fluoresces more brightly after uptake into BLECs.

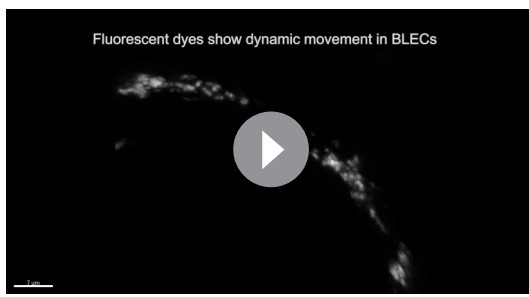

**Video 4.** Fluorescent dyes show dynamic movement in BLECs. Confocal projection of a one-hour video shows tracer movement in BLECs in 5dpf zebrafish head after injection of fluorescent dyes into the center of TeO close to the meningeal surface as shown in *Figure 7A*. IgG-647 tracer exhibits dynamic movements in two neighboring BLECs. Image stacks were recorded every two minutes. Data is representative of at least five independent experiments. BLEC, brain lymphatic endothelial cell; dpf, days post fertilization; IgG-647, IgG-conjugated Alexa Fluor 674; TeO, Optic Tectum.

of acidic vacuoles seen in platelets (*Hernandez-Ruiz et al., 2007*) but also secretory granules seen in mast cells (*Wernersson and Pejler, 2014*). The presence of large inclusions suggests that BLECs are possibly similar to Mato-cells in rodents (*Mato et al., 1984*), which reside in the perivascular space between the basal lamina of blood vessels and the glia limitans of the neuropil, the so-called Virchow-Robin space. Due to the high content of fluorescent, endocytic granules, Mato cells are also referred to as fluorescent granular perithelial cells (FGP) cells (*Mato et al., 1985*). However, Mato/FGP cells are typically described to contain a more heterogeneous population of randomly clustered, electron dense and light inclusions that often become foamy and darker with age and under pathological conditions (*Mato et al., 2009*, *Mato et al., 1997*). In some images (*Figure 6A*) we observe cells with heterogeneous inclusions typical of FGP cells, but which appear distinct from BLEC cells. We therefore favor the notion that BLECs and FGP cells are distinct cell populations.

The presence of large spherical vacuoles caused us to investigate whether BLECs can clear macromolecules during embryonic stages until 5dpf. We found that BLECs rapidly take up exogenous molecules into small cytoplasmic vesicles. These vesicles also readily accumulate the pH-sensitive dye pHr, suggesting intracellular localization to low pH lysosomes (*Figure 7H*). In addition, our photoconversion data using Kaede protein suggest a dynamic process of uptake and clearance of internal-

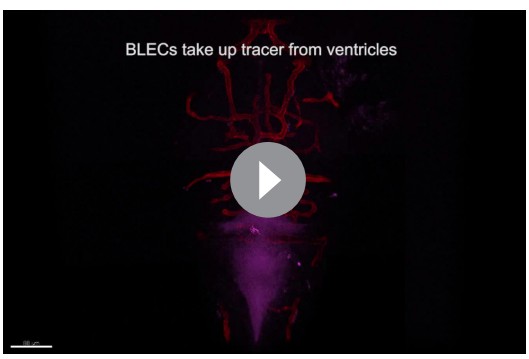

**Video 5.** BLECs take up tracer from ventricles. Complete 2.5 hr time-lapse of dorsal confocal projections of the zebrafish head region in a 72hpf embryo after IgG-647 ICV injection as depicted in *Figure 7—figure supplement 3*. Blood vessels are highlighted in red (*kdr-l:mCherry*) and lympho-venous cells in green (*flt4:mCitrine*). IgG-647 is pseudo-colored in purple. Note uptake of tracer into *flt4* positive BLECs over time. Image stacks were recorded every five minutes. Data are representative of at least three independent experiments. BLEC, brain lymphatic endothelial cell; hpf, hours post fertilization; IgG-647, IgG-conjugated Alexa Fluor 674.

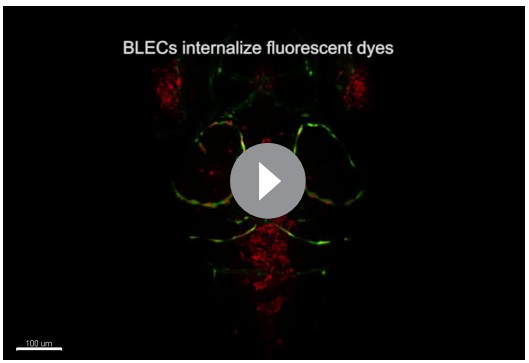

**Video 6.** BLECs internalize fluorescent dyes. 3D animation of a dorsal confocal projection of the zebrafish head region at 5dpf injected with pHr into the TeO as depicted in *Figure 7A*. Lympho-venous cells are in green (*flt4:mCitrine*), pHr in red. BLECs strongly label with the pH-sensitive dye pHrodo (pHr) indicating tracer internalization. Signal from eye and skin pigmentation is auto-fluorescence. Origin of dot-like pHr signal not associated to BLECs is unknown but may represent tracer aggregation. Data is representative of at least five independent experiments. BLEC, brain lymphatic endothelial cell; dpf, days post fertilization; TeO, Optic Tectum; pHr, pHrodo Red Avidin.

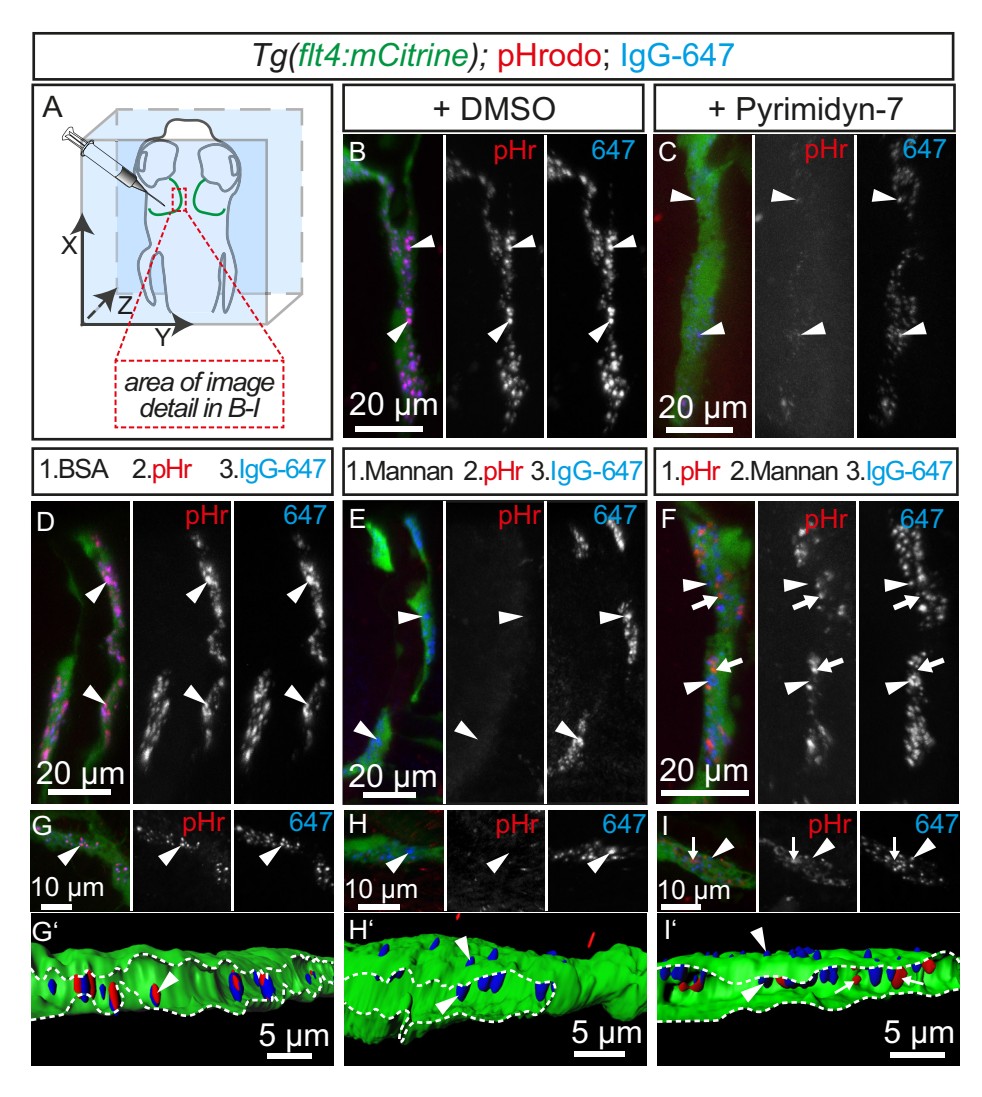

**Figure 8.** Tracer uptake by BLECs is inhibited by mannan administration. (**A**) Overview of the zebrafish head region depicting intratectal injection of fluorescent dyes into the center of the TeO close to the meninges in 5dpf embryos. Red inset denotes area of image detail for representative dorsal confocal projections in **B–I**. (**B**) and (**C**) In DMSO treated fish co-injection of pHr and IgG-647 reveals high degree co-localization of both tracers in vesicles within mCitrine positive BLECs (**B**, white arrowheads). Inhibition of dynamin-dependent endocytosis with Pyrimidyn-7 results in reduced signal intensity from IgG-647 positive vesicles and an almost complete block of pHr uptake (**C**, white arrowheads). (**D–F**) Separate, consecutive injections of BSA or mannan with pHr and IgG-647. Injection of BSA, pHr and IgG-647 results in co-localization of both tracers in BLECs (**D**, white arrowheads). In contrast, initial injection of mannan completely blocks the uptake of pHr but not the accumulation of IgG-647 to BLECs (**E**, white arrowheads). However, initial injection of pHr followed by mannan and IgG-647 results in accumulation of both tracers in BLECs with hardly any co-localization (**F**, white arrowheads and arrows). (**G–I**) Same experimental setup as in **D–F** except the embryos were fixed in PFA five minutes after the last injection to enable higher resolution imaging. Area within the white dotted lines in 3D-reconstructions reveals the lumen of BLECs (**G'**,**H'** and **I'**). In BSA control injected fish both dyes co-localize in intraluminal vesicles (**G** and **G'**, white arrowheads). In mannan injected fish, IgG-647 cannot be internalized completely but remains stuck at the cell surface, (**H** and **H'**, white arrowheads). Finally, initial injection of pHr reveals internalized vesicular pHr and membrane associated IgG-647 (**I** and **I'**, white arrows and white arrowheads). Data are representative of at least two independent experiments. BLEC, brain lymphatic endothelial cell; BSA, bovine serum albumin; dpf, days post fertilization; IgG-647, IgG-conjugated Alexa Fluor 674; PFA, paraformaldehyde; pHr, pHrodo Red Avidin; TeO, Optic Tectum.

*Figure 8 continued on next page*

*Figure 8 continued*

The following figure supplements are available for figure 8:

**Figure supplement 1.** BLECs express *mrc1a*.
**Figure supplement 2.** Tracer uptake by BLECs is not inhibited by galactose administration.

ized cargo (*Figure 7I–J*). Taken together, we conclude that BLECs remove extracellular molecules by continuous internalization and subsequent degradation in lysosomes. As mentioned earlier, this process resembles the activity of macrophages or dendritic cells, which selectively internalize substances through receptor-mediated endocytosis (RME) (*Roche and Furuta, 2015*). Indeed, the selectivity for small and intermediate size macromolecules indicates specific uptake through RME in BLECs (*Figure 7C–E*), likely involving the MR, which is expressed in BLECs but was not detected in neighboring blood vessels (*Figure 8—figure supplement 2*) (*Taylor et al., 2005*). After cargo binding and internalization, the MR unloads bound cargo for degradation in lysosomes. Since the MR continuously recycles between the cytosol and the PM, it enables rapid uptake of exogenous molecules (*Martinez-Pomares, 2012*). BLECs show hallmarks of MR-mediated uptake including the endocytosis of fluorescently labelled dextran (*Figure 7D*) and sensitivity to the inhibition of dynamin, a small GTPase essential for RME in eukaryotic cells (*Martinez-Pomares, 2012*; *McMahon and Boucrot, 2011*) (*Figure 8C*). Finally, RME is a saturable process and we show complete inhibition of pHr uptake after administration of the competitive agonist mannan (*Sallusto et al., 1995*) (*Figure 8E*), but not after administration of galactose (*Figure 8—figure supplement 2*). Therefore, we suggest MR-dependent RME as a major entry portal for glycoproteins and polysaccharides into BLECs. To our knowledge, this is the first study to describe the intracellular uptake and clearance of macromolecules by individual LECs.

In mice, MR functions in the maintenance of macromolecule homeostasis in the blood and MR loss-of-function results in reduced clearance of Lutropin (LH) (*Mi et al., 2002*), denatured collagens (*Malovic et al., 2007*) and serum glycoproteins including most lysosomal hydrolases (*Lee et al., 2002*). The latter are upregulated during inflammation and their MR-mediated removal may therefore prevent excess tissue damage. Although not addressed experimentally, MR expressing LSCEs likely represent the cell type involved in the removal of serum glycoproteins, analogous to the clearance of collagens by LSCEs (*Malovic et al., 2007*). An intriguing hypothesis is that BLECs may play a functionally similar role in the maintenance of cerebrospinal fluid (CSF) homeostasis through MR-mediated removal of potentially noxious molecules such as lysosomal enzymes. Indeed tracer injected into the CSF rapidly accumulates in BLECs (*Figure 7—figure supplement 3*, *Video 5*). Such a mechanism may be important in disease involving increased deposition of lysosomal enzymes into the CSF, which in humans is a hallmark of bacterial meningitis (*Beratis et al., 1997*). MR-mediated endocytosis by macrophages and dendritic cells has also been involved with pathogen recognition and antigen presentation during adaptive immunity. However, although surface glycoproteins from various pathogens internalize through MR, the analysis of MR null mice did not reveal increased sensitivity to infection (*Taylor et al., 2005*). Further, our data suggest that brain resident *mpeg1* positive macrophages do not express *mrc1a*, which was restricted to BLECs in the meninges (*Figure 8—figure supplement 1*) and we did not observe apparent tracer uptake in cells other than BLECs in their immediate surroundings. Similarly, MR was not found to be involved in FITC uptake by zebrafish kidney phagocytes (*Hohn et al., 2009*). Therefore, it is plausible that BLECs in zebrafish may exert functional roles in macromolecule uptake and pathogen recognition normally carried out by macrophages and dendritic cells in mammals.

Our data show that even in the presence of mannan, IgG-647 can adhere to the cell surface of BLECs (*Figure 8E,F,H,I*). This may be explained by the involvement of a second receptor with similar ligand binding properties as the MR that requires binding to the MR for internalization. For instance, macrophages, sinusoidal ECs and human skin LECs express the stabilin class of scavenger receptors for endocytosis and degradation of various non-self antigens (*Kzhyshkowska et al., 2006*), and *stabilin1* is expressed in lymphatic endothelial cells in zebrafish (*Hogan et al., 2009a*). Moreover, another PRR, Toll-like receptor 2 (TLR2) forms a complex with MR on the cell surface in response to

pathogen interaction in alveolar macrophages (*Tachado et al., 2007*). Interestingly, human LECs isolated from different tissues differentially express various TLRs including TLR2 (*Garrafa et al., 2011*).

Another remaining question is whether BLECs are involved in phagocytosis of invading pathogens. The large vacuoles in BLECs, which we observed only in adult fish but not in larval stages, may also represent phagosomes rather than endocytic vesicles (*Figure 5G*). Interestingly, phagocytosis has been shown to be independent of MR activity in mouse macrophages (*Lee et al., 2003*). Therefore, certainty about the receptors involved in macromolecule uptake in BLECs as well as in how far these cells assume the classical role of macrophages or dendritic cells awaits further investigation.

In future studies, it will also be important to determine whether zebrafish BLECs participate in glymphatic-paravascular clearance of solutes and waste products from the energy intensive brain, as has been a proposed function for murine meningeal lymphatics (*Aspelund et al., 2015*; *Louveau et al., 2015*). In rodents, multiple routes of solute clearance from the brain have been proposed to run either along arterial routes in paravascular compartments (*Rennels et al., 1985*; *Iliff et al., 2013*) or more specifically along peri-vascular basement membranes of capillaries and arteries (*Weller et al., 2008*). It remains to be elucidated how these routes are interconnected, and how they interact with the lymphatic vasculature of the rodent meninges (*Aspelund et al., 2015*; *Louveau et al., 2015*). If BLECs are functionally related to rodent LECs, the optical transparency of the zebrafish promises to allow for direct observation of these processes in vivo. Moreover, there is growing evidence that dysfunction of these clearance mechanisms may trigger or exacerbate neurodegenerative disease. For example, amyloid beta (A$\beta$) has been shown to aggregate and move along basement membranes of capillaries and arteries in the brain (*Weller et al., 2008*), and Cerebral Amyloid Angiopathy can occur when this pathway is disrupted (*Preston et al., 2003*; *Weller et al., 2009*). Other risk factors for Alzheimer's Disease and dementia, including the cholesterol transporter isoform ApoE4 (*Liu et al., 2013*), age-related deterioration of arterial integrity and disruption of elasticity (*Weller et al., 2009*), or changes to CSF production (*Silverberg et al., 2001*, *Silverberg et al., 2003*), have been implicated in the disruption of perivascular clearance, specifically including A$\beta$ clearance (*Hawkes et al., 2012*). Dysfunctional solute clearance from the CSF surrounding the optic nerve has also been suggested to alter interocular pressure contributing to the onset of glaucoma (*Wostyn et al., 2015*). Interestingly, sleep and anesthesia also directly impact clearance by opening paravascular compartments and increasing tracer influx form the CSF into the brain (*Xie et al., 2013*), indicating that sleep disruption may play a critical role in the development and exacerbation of neurodegenrative pathologies (*Musiek and Holtzman, 2016*). How zebrafish brain states and aging influence BLEC macromolecule uptake will be important questions for future studies, paving the way for small molecule screens for regulators of this clinically relevant process (*Rihel et al., 2010*).

We demonstrate here an endothelial cell population that emerges from a venous blood vessel in the zebrafish embryonic brain and exhibits many critical hallmarks of lymphatic identity, such as *prox1a*, *lyve1* and *vegfr3* expression. These cells share properties with, but appear to be distinct from, macrophages and combine structural and molecular features of endothelial cells with functional characteristics of macrophages and microglial cells. Intriguingly, these brain lymphatic endothelial cells persist through larval and adult stages and are situated in the adult meninges, in close proximity to blood vessels. A comparison of zebrafish BLECs with murine meningeal LECs should provide interesting features of conserved and divergent molecular signatures, cellular functions and physiological relevance. As the mammalian glymphatic-paravascular clearance mechanism has recently received significant attention and recognition as a vascular entity involved in physiological and pathophysiological conditions such as behavior, sleep, waste removal, Alzheimer's disease, and others, the identification of BLECs provides an additional system to experimentally address how lymphatic endothelial cells contribute to brain function.

## Materials and methods

### Zebrafish strains

Animal work followed guidelines of the animal ethics committees at the University of Münster, Germany, and the University College London, England. The following transgenic and mutant lines have been used in this study:

*Tg(kdr-l:HRAS-mCherry-CAAX)$^{s916}$* (**Hogan et al., 2009a**); *Tg(lyve1:dsRed2)$^{nz101}$* (**Okuda et al., 2012**), *Tg(flt4$^{BAC}$:mCitrine)$^{hu7135}$* and *Tg(prox1a$^{BAC}$:KalTA4-4xUAS-E1b:uncTagRFP)$^{nim5}$* herein denoted *Tg(flt4:mCitrine)* and *Tg(prox1a:KalTA4-UAS:TagRFP)* (**van Impel et al., 2014**), *Tg(flt1$^{enh}$: tdTomato)$^{hu5333}$* (**Bussmann et al., 2010**), *Tg(mpeg1:EGFP)$^{gl22}$* (**Ellett et al., 2011**) and *ccbe1$^{hu10965}$* (**Kok et al., 2015**).

Embryos, larval and adult zebrafish (Danio rerio) were kept either at the UK Münster (embryos at 28.5°C and adults at 26°C with a 12 hr light and 12 hr dark cycle) or at University College London's fish facility (at 28°C with a 14 hr light and 10 hr dark cycle). Experimental procedures were conducted under the project licence awarded to J.R. from the UK Home Office, according to the UK Animals (Scientific Procedures) Act 1986.

## Genotyping

DNA was isolated as described previously (**Dahlem et al., 2012**). Genotyping for *ccbe1$^{hu10965}$* mutants was performed by PCR amplification (**Table 1**) followed by restriction digest with SacI (Promega, # R6061). Analysis was performed by standard gel electrophoresis.

## Live imaging

Live imaging was carried out with 3dpf embryos up to 7 mm larvae. Transgenic strains were kept in a wild-type (AB) or *nacre (mitfa)$^{w2/w2}$* mutant background, which has reduced pigmentation (**Lister et al., 1999**). 1-Phenyl-2-thiourea (PTU, 75 μM, Sigma, #P7629) was added before 24 hr post fertilization (hpf) to inhibit melanogenesis (**Karlsson et al., 2001**). For imaging, embryos were anesthetized with 42 mg/L MS222 (Sigma, #A5040) and embedded in 1% low melting agarose (Thermo-Fischer, #16520100) dissolved in embryo medium. Embryo medium containing MS222 was layered on top of the agarose once solidified. For imaging of larvae, the agarose was removed around the head region.

## Injection regimes

Injections were carried out with a Pneumatic PicoPump (WPI, #PV 820) and glass capillary needles (Science Products Gmbh, # GB100TF-10) prepared with a Micropipette Puller (Shutter Instruments, # P-1000). 4 ng of *pU.1* morpholinos were injected as previously described (**Rhodes et al., 2005**).

For all other injections, embryos were anesthetized in 2% low melting agarose (ThermoFischer, #16520100) dissolved in embryo medium containing 42 mg/L MS222 (Sigma, #A5040) and injected with a total volume of 0.5 nl - 1 nl per injected bolus. For intratectal injection, needles were inserted into the brain in a sloped angle into the center of the optic tectum. Care was taken not to penetrate deep into the brain tissue and the injection bolus was unloaded close to the meninges. Injection into the hindbrain (rhomboencephalic) ventricle was carried out as described before (**Fame et al., 2016**).

The following fluorescent dyes and concentrations were used for injection: 10 kDa dextran-conjugated Alexa Fluor 647 (2mg/ml, ThermoFischer, #D22914), 500 kDa dextran-conjugated Cy5 (10 mg/ml, Nanocs, #DX500-S5-1), 2000 kDa dextran-conjugated FITC (3 mg/ml, Sigma, #FD2000S), Evans Blue (1% in PBS, Sigma, #E2129), IgG-conjugated Alexa Fluor 674 (2mg/ml, ThermoFischer, #A31573, RRID:AB_2536183), pHrodo Red Avidin (2 mg/ml, ThermoFischer, #P35362). Kaede protein (2 mg/ml) was a generous gift from B. Feldmann (**Brown et al., 2008**).

## Inhibition of endocytosis

For inhibition of dynamin dependent endocytosis, embryos were transferred to embryo medium containing DMSO control or 30 μM Pyrimidyn-7 (30 μM, Abcam, #ab144501) five minutes prior to dye injection. For inhibition of the mannose receptor, embryos were injected with 0.5–1 nl BSA (50

**Table 1.** Primers used for genotyping *ccbe1$^{hu10965}$* embryos.

| Primer name | Sequence |
| --- | --- |
| ccbe1_TALEN_Control-FW | GAACCTATGGAAGCCGATCA |
| ccbe1_TALEN_Control-Rv | GCCTACAGACAATACACAAACACA |

mg/ml, Sigma, A2153) or D-(+)-Galactose (50 mg/ml, Sigma, G0750) or mannan (50 mg/ml, Sigma, #M7504) into the optic tectum five minutes prior to dye injection.

## Immunohistochemistry

### Whole mount stainings

Immunofluorescent antibody labeling in embryonic whole mounts was performed as described previously with minor modifications (*Koltowska et al., 2015*; *Karpanen et al., 2017*). In brief, 3dpf embryos were fixed overnight in 4% PFA-PBS at 4°C, washed 3x in ice-cold MeOH, and then incubated 1 hr in 3% $H_2O_2$ in MeOH on ice. Embryos were washed 3x in ice-cold MeOH and kept in MeOH at −20°C for 2 days. Next embryos were washed 3x in PBS-T (0.1% Tween 20 in PBS) and then cryoprotected overnight in PBS-T containing 30% sucrose at 4°C. After 3x washes in PBS-T, embryos were transferred to 150 mM Tris-HCl (pH 9.0) for 5 min, heated to 70°C for 15 min and then incubated in ice-cold acetone at −20°C for 40 min. Following 2x washes in PBS-T, embryos were treated with Proteinase K (10 µg/ml in PBS-T) for 30 min and then re-fixed in 4% PFA-PBS for 20 min. Then embryos were washed 3x in PBS-T and 1x in TBST (0.1% Triton X-100 in TBS) and incubated overnight in blocking buffer (1% BSA, 10% goat serum in TBST) at 4°C. Next, primary antibodies were incubated in blocking buffer overnight at 4°C. Embryos were washed 5x in TBST and 1x in Maleic buffer (150 mM Maleic acid, 100 mM NaCl, 0.001%Tween20, pH7.4) for 30 min and then blocked in Maleic buffer containing 2% Blocking Reagent (Roche, #11096176001) for 2 hr. Secondary antibodies were incubated overnight in Maleic buffer containing 2% Blocking Reagent at 4°C. Embryos were washed 5x in Maleic buffer and 1x in PBS. Finally, embryos were in 150 µL TSA Plus Cyanine 3 System reagent (PerkinElmer, #NEL744001KT) for 3 hr in the dark. Embryos were washed 2 days in TBST and embedded for imaging. Primary antibodies used were α-Prox1 (1:500, AngioBio, #11–002) and α-GFP (1:400, Abcam, #ab13970). Secondary antibodies used were HRP-Goat α-Rabbit IgG (1:1000, ThermoFischer,T20922) and Goat α-Chicken Alexa Fluor 488 (1:200, ThermoFischer, #A11039.).

### Whole mount stainings of adult fish brains

Dissected adult brains from four month old *Tg(flt4:mCitrine); Tg(kdr-l:mCherry)* fish were labelled as whole mounts following standard procedure (*Turner et al., 2016*). Primary antibodies used were α-RFP (1:500, Rockland USA, #600-901-379), α-RFP (1:100, MBL, #PM005), α-GFP (1:1000, Nacalai Tesque, 04404–84). Secondary antibodies used were α-chicken Alexa Fluor 568 (1:2000, ThermoFischer, #A11041), α-rat Alexa Fluor 488 (1:2000, ThermoFischer, A11006). DAPI (1:1000, ThermoFischer, D1306).

### Cryosections of adult fish brain

Coronal cryosections of adult brains from 14 month old *Tg(flt4:mCitrine); Tg(kdr-l:mCherry)* fish were labelled as per (*Turner et al., 2016*) with the following amendments: adult brains were frozen via a dry ice / ethanol bath and cut in 20micron sections. A mixture of secondary antibodies in blocking serum (10% normal goat serum, 1% DMSO, 0.5% Triton x-100 in PBS) were added and incubated at room temperature for 3.5 hr, then washed 3 × 10 min in the dark. Glass slides were then coverslipped using a mountant media (Citifluor, UK) and stored at 4°C in the dark. Primary antibodies used were α-RFP (1:2000, Rockland USA, #600-901-379), α-GFP (1:2000, Nacalai Tesque, 04404–84). Secondary antibodies used were α-chicken Alexa Fluor 568 (1:2000, ThermoFischer, #A11041), α-rat Alexa Fluor 488 (1:2000, ThermoFischer, A11006). DAPI (1:1000, ThermoFischer, D1306).

## Whole-mount in situ hybridization

Anti-sense RNA probes were generated from cDNA using direct amplification and transcription with T3 RNA polymerase. Single in situ hybridizations were carried out as described previously (*Schulte-Merker, 2002*; *Thisse and Thisse, 2008*). For the detection of *mrc1a* 100 ng of the probes from primer pair 1 and 2 (*Table 2*) were combined and hybridized as described previously to *Tg(flt4:mCitrine)* fish with a *nacre (mitfa)^(w2/w2)* mutant background.

Double fluorescent in situ hybridizations were adapted from (*Brend and Holley, 2009*) with the following amendments: dissected 7dpf brains were digested with proteinase K (5 mg/mL in PBST) for 5 min at room temperature. Pre-hybridization was carried out with full Hybridization+ (Hyb+)

**Table 2.** Primers used for synthesis of ISH probes.

| Primer pair | Gene / Region | Sequence | Probe lengths |
|---|---|---|---|
| Pair 1: Forward primer | mrc1a/3'UTR | CTAGGCCTGCGATTGGAGAG | 547bps |
| Pair 1: Reverse primer | mrc1a/3'UTR | CATTAACCCTCACTAAAGGGAACTGCCACCTCATGTCCAGTT | |
| Pair 2: Forward primer | mrc1a / Exon 2 | ACACCAGCTACTTCCTTATCTACA | 401bps |
| Pair 2: Reverse primer | mrc1a / Exon 2 | CATTAACCCTCACTAAAGGGAATTGATTTCCATGAGAACATAAATCGT | |
| Pair 3: Forward primer | mrc1a/3'UTR | CCACAGACATAGCTCCCACA | 987 bp |
| Pair 3: Reverse primer | mrc1a/3'UTR | GAAATAATTAACCCTCACTAAAGGGACCAGCAGTCTATTTGGCTATTC | |

solution added to embryos and incubated overnight at 67°C. 100 ng of probe from primer pair 3 (*Table 2*) was then added to fresh Hyb+ and incubated overnight at 67°C. After stringency washes, 10% NGS/PBS (Block) was added for 1 hr at room temperature, followed with anti-dig POD antibody (1:1000, Roche #11207733910 TSA Cyanine 3 Tyramide Reagent Pack) added in Block overnight at 4°C. Staining was via Cyanine 3 diluted 1:50 in amplification diluent (Perkin Elmer, #SAT704A001EA) for 60 min at room temperature. Primary antibodies used were chicken α-GFP (1:500, Abcam, #ab13970) and rabbit α-GFP (1:1000, Amsbio, #TP401) both applied overnight at 4°C. Secondary fluorescent antibodies goat α–Chicken IgG 488 (1:200, Lifetech, #A-11039) and goat α-Rabbit 488 (1:200, Lifetech, #A-11034) were both applied overnight at 4°C. The next day embryos in secondary antibody were removed from 4°C and placed on a rocker at room temperature for 2 hr before final washes and imaging.

## Transmission electron microscopy

Terminally anaesthetized adult zebrafish were heart perfused with room temperature EM fixative (2% Gluteraldehyde, 2% Paraformaldehyde buffered with 0.1M sodium cacodylate pH7.3) and placed in the same fixative overnight at 4°C. The brain was dissected out, split into brain regions and osmocated (1% Osmium tetroxide, buffered with 0.1M sodium cacodylate) followed by dehydration in an ethanol series, transfer to propylene oxide, infiltration with epoxy resin overnight ('medium' agar 100, Agar scientific UK), and polymerization for 24 hr at 60°C. Semi and ultrathin sections were cut and stained with uranyl acetate and lead citrate before observation and photography using a Phillips TEM (MODEL,KeV).

## Microscopy and image processing

Images of live embryos, embryonic fish and adult brain whole mounts were imaged with a Leica SP8 microscope using 10x or 20x dry objectives and 25x and 40x water immersion objectives. Scoring of *ccbe1* mutant embryos (*Figure 4E*) was performed using a Leica M165 FC and an X-Cite 200DC (Lumen Dynamics) fluorescent light source. Cryosections were imaged with a Zeiss Airyscan 880 using a 63x oil immersion objective. In situ hybridizations (ISH) were imaged with a Nikon Eclipse Ni using 10x and 20x objectives. Brains from double fluorescent ISH were imaged on a Leica SPE with 20x objective. Confocal stacks were processed using Fiji-ImageJ version 1.51g or Imaris version 7.7.2. Images and figures were assembled using Microsoft Power Point and Adobe Photoshop and Adobe Illustrator. All data was processed using raw images with brightness, colour and contrast adjusted for printing.

## Acknowledgements

We thank members of the Schulte-Merker, Rihel, Wilson, and Bianco labs for discussions, D Stainier for providing transgenic fish lines, M Turmaine for his assistance with EM and UCL's fish facility team for their excellent work. S Reichert and J Lau assisted with graphical design, L Goodings provided technical expertise. B Feldmann (NIH; Bethesda) and F Peri (EMBL, Heidelberg) generously provided Kaede protein and pU.1 morpholino, respectively. The work was supported by the DFG (SCHU 1228/2–1, Forschergruppe FOR2325, Interactions at the Neurovascular Interface) and the CiM

Cluster of Excellence (EXC 1003 CiM, WWU Münster, Germany), and a UCL Excellence Fellowship and European Research Council Starting grant awarded to JR.

## Additional information

### Funding

| Funder | Grant reference number | Author |
| --- | --- | --- |
| Deutsche Forschungsgemeinschaft | FOR2325 | Stefan Schulte-Merker |
| Deutsche Forschungsgemeinschaft | CiM 1003 | Max van Lessen |

The funders had no role in study design, data collection and interpretation, or the decision to submit the work for publication.

### Author contributions

MvL, SS-G, Conceptualization, Data curation, Formal analysis, Investigation, Visualization, Writing—original draft, Writing—review and editing; AvI, Formal analysis, Investigation, Visualization, Writing—review and editing; TAH, Conceptualization, Formal analysis, Investigation, Writing—original draft, Writing—review and editing; JR, Conceptualization, Supervision, Funding acquisition, Investigation, Visualization, Writing—original draft, Writing—review and editing; SS-M, Conceptualization, Formal analysis, Supervision, Funding acquisition, Validation, Methodology, Writing—original draft, Writing—review and editing

### Author ORCIDs

Shannon Shibata-Germanos, http://orcid.org/0000-0002-4250-7497
Andreas van Impel, http://orcid.org/0000-0002-4737-3547
Jason Rihel, http://orcid.org/0000-0003-4067-2066
Stefan Schulte-Merker, http://orcid.org/0000-0003-3617-8807

### Ethics

Animal experimentation: Experimental procedures were conducted under project licence awarded to J.R. from the UK Home Office (Permit Number: 70/7612), according to the UK Animals (Scientific Procedures) Act 1986.

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
