## [Decision Letter]

Thank you for submitting your article "Intracellular uptake of macromolecules by brain lymphatic endothelial cells during embryonic development" for consideration by *eLife*. Your article has been favorably evaluated by Didier Stainier (Senior Editor) and three reviewers, one of whom, Holger Gerhardt (Reviewer #1), is a member of our Board of Reviewing Editors.

The reviewers have discussed the reviews with one another and the Reviewing Editor has drafted this decision to help you prepare a revised submission.

Summary:

The reviewers find the work presents a highly original and interesting finding, demonstrating for the first time that also zebrafish possess brain lymphatics. Intriguingly, the identified cells clearly share molecular markers and pathway dependency with lymphatic vessels elsewhere in the zebrafish or in other vertebrate species. However, instead of forming patent lumenized and fluid draining vessels, they form a network of loosely connected cells displaying functional properties of perivascular macrophages. Elegant tracer studies demonstrate phagocytic activity, involvement of the mannose receptor and endocytic uptake and acidification. The reviewers find the work very well written and beautifully illustrated, and agree that most of the conclusions are sufficiently substantiated.

However, two points were highlighted as deserving further clarification before the work can be accepted for publication. The first relates to the claim of mannose receptor function in endocytosis, the second concerns the relationship between the brain lymphatic cells and blood vessels.

Essential revisions:

1) The reviewers request that you demonstrate directly that the BLEC express the mannose receptor, and that you comment on the possible nature of physiologically phagocytised cargo. The artificial cargo used in your study demonstrates phagocytic activity, but not what these cells might take up normally.

2) The reviewers felt it would be important to comment on the relationship between BLEC and blood vessels in the brain. Is the attraction to the blood vessels mediated by localised Vegfc and ccbe expression, and does the lack of the cells in the mutants affect the blood vessels?

The question of mannose receptor expression will likely need to be addressed experimentally, and the expression domain of Vegfc may also require labelling. However, we hope this may be accomplished with a short turn-around time and shouldn't significantly delay revision.

---

## [Author Response]

*Essential revisions:*

*1) The reviewers request that you demonstrate directly that the BLEC express the mannose receptor, and that you comment on the possible nature of physiologically phagocytised cargo. The artificial cargo used in your study demonstrates phagocytic activity, but not what these cells might take up normally.*

We have carried out ISHs and now demonstrate expression of mannose receptor 1a (mrc1a) in BLECs. These data have been added as Figure 8—figure supplement 1.

We have also now included comments on the possible nature of phagocytised cargo under physiological conditions (see Discussion, eighth paragraph).

*2) The reviewers felt it would be important to comment on the relationship between BLEC and blood vessels in the brain. Is the attraction to the blood vessels mediated by localised Vegfc and ccbe expression, and does the lack of the cells in the mutants affect the blood vessels?*

We have addressed this question in two ways. First, during early development (56-72hpf), we carried out ISHs for vegfc and ccbe1, in order to examine whether expression of either gene might directly explain the outgrowth of BLECs in the vicinity of the MsV. Five independent experiments in two different labs were unable to show expression to be enriched in the MSV area. We have included a sentence in the Results section (see subsection “Sprouting endothelial cells express Prox1 and are sensitive to genetic ablation of 109 ccbe1 but not pU.1”, third paragraph).

Second, we have included two sentences in the Discussion section to clarify the relationship between blood vessels and BLECs in later stages (such as depicted in Figure 5). Specifically, we cite the work by Bower et al., (Bower et al., 2017) and Venero Glanternik (Venero Glanternik et al., 2017) which show high levels of vegfab and egfl7 expression in BLECs, suggesting that blood vessels align along BLECs, and not vice versa (see Discussion, fifth paragraph). Furthermore, Bower et al., (Bower et al., 2017) have demonstrated that in a laser ablation model, blood vessels regenerate in the larval brain, likely using BLECs as guiding cells during regeneration. Hence, BLECs might be essential for meningeal blood vessel formation to occur in vivo, and we discuss this in the revised version of our manuscript (see Discussion, fifth paragraph).